# MiR-155 promotes compensatory lung growth by inhibiting JARID2 activation of CD34+ endothelial progenitor cells

Li Zhao[1], Jing Peng[1], Li Zhuang[2], Zhiling Yan[3], Fei Liao[4], Yifan Wang[1], Shihao Shao[1], Weiwei Wang[5]*

1 Department of Anesthesiology, Yunnan Cancer Hospital, The Third Affiliated Hospital of Kunming Medical University, Kunming, 650118, Yunnan, China, 2 Department of Palliative Medicine, Yunnan Cancer Hospital, The Third Affiliated Hospital of Kunming Medical University, Kunming, 650118, Yunnan, China, 3 Department of Gynaecologic Oncology, Yunnan Cancer Hospital, The Third Affiliated Hospital of Kunming Medical University, Kunming, 650118, Yunnan, China, 4 Department of Anesthesiology, The 6th Affiliated Hospital of Kunming Medical University (The People's Hospital of Yuxi City), Yuxi, 653100, Yunnan, China, 5 Department of Thoracic Surgery II, Yunnan Cancer Hospital, The Third Affiliated Hospital of Kunming Medical University, Kunming, 650118, Yunnan, China

☯ These authors contributed equally to this work.
* wangweiweiqj@126.com

**Data Availability Statement:** I have uploaded my data to the figshare database and below is a link to the data. link: https://figshare.com/s/cbde4268cf2ae018b832.

## Abstract

Bone marrow-derived CD34-positive (CD34+) endothelial progenitor cells (EPCs) has unique functions in the mechanism of compensatory lung growth (CLG). The content of this study is mainly to describe the effect of microRNA (miR)-155 in the mechanisms of EPCs and CLG. Our study found that transfection of miR-155 mimic could promote EPC proliferation, migration and tube formation, while transfection of miR-155 inhibitor had the opposite effect. It was also found that transfection of pc-JARID2 inhibited EPC proliferation, migration and tube formation, while transfection of si-JARID2 had the opposite effect. miR-155 can target and negatively regulate JARID2 expression. Overexpression of JARID2 weakened the promoting effects of miR-155 mimic on EPC proliferation, migration, and tubular formation, while silencing JARID2 weakened the inhibitory effects of miR-155 inhibitors on EPC proliferation, migration, and tubular formation. Transplantation of EPCs transfected with miR-155 mimic into the left lung model effectively increased lung volume, total alveolar number, diaphragm surface area, and lung endothelial cell number, while transplantation of EPCs co-transfected with miR-155 mimic and pc-JARID2 reversed this phenomenon. Overall, we found that miR-155 activates CD34+ EPC by targeting negative regulation of JARID2 and promotes CLG.

## 1 Introduction

In most mammals, the removal of one lung leads to rapid growth of the remaining lung mass is called compensatory lung growth (CLG) [1,2]. According to a large number of literatures, this phenomenon has occurred in mice [3], rats [4], pigs [5], dogs [6], rabbits [7] and ferrets

**Funding:** This study was supported by Yunnan Provincial Science and Technology Department-Kunming Medical University (Grant No.202201AY070001-137); Joint Special Fund for Applied Basic Research of Yunnan Provincial Science and Technology Department-Kunming Medical University (No. 202101AY070001-170, No. 202301AY070001-085).

**Competing interests:** The authors declare that they have no competing interests.

[8], the mechanism for the occurrence of CLG is still puzzling. CLG effect triggered after pneumonectomy can lead to increased lung volume and alveolar number, and they showed that it is a key mechanism for neovascularization [9]. It has recently been shown that within 3 weeks after pneumonectomy, the number of endothelial cells in the lungs of mice increased 3.2 times and the number of CD34$^+$ endothelial cells increased 7.3 times in time. Blood-derived CD34$^+$ endothelial progenitor cells (EPCs) become resident endothelial cells during CLG and contribute to pulmonary angiogenesis [10]. Therefore, we speculate that the mechanism of CLG may be related to the development of EPCs.

EPCs are bone marrow stem cells, which have the ability to differentiate into mature cells, promote the formation of new blood vessels, and maintain the integrity of blood vessels [11,12]. Studies have shown that microRNAs (miRNAs) can regulate EPCs proliferation, migration and angiogenesis [13,14]. miRNAs are relatively small in molecular weight and are 18–23 nucleotides in length. They can silence or degrade mRNA, and then metabolize and degrade mRNA in biological processes. regulation of death [15]. After consulting the literature, we know that MiR-155 is inextricably linked with the proliferation and migration of CD34$^+$ cells. In the study of inducing adipose-derived stem cells (ASCs) to differentiate into endothelial cells, miR-155 can mediate the occurrence of endothelial differentiation of CD34+ ASCs [16]. To explore whether miR-155 can make CD34$^+$ chronic myeloid leukemia (CML) cells eliminate the growth inhibition of TGF-β1 and signal transduction of bone morphogenetic protein (BMP), miR-155 was overexpressed, which significantly promoted the proliferation rate of CD34$^+$ CML cells [17]. We speculate that the proliferation of CD34$^+$ endothelial cells in CLG can also be regulated by miR-155.

MiRNA regulates gene expression at the posttranscriptional level mainly through complementary pairing with target mRNA [18], and software prediction and literature showed that JARID2 and miR-155 had a targeted binding relationship [19–22]. JARID2 is a member of the Jumonji family of proteins that have been shown to regulate cell proliferation and migration [23]. In primary myofibrosis (PMF) studies, JARID2 silencing has been shown to promote proliferation and differentiation of normal CD34$^+$ cells [19]. In addition, in the study of the effects of JARID2 on trophoblast cell viability and invasion, it was also found that JARID2 silencing could significantly reduce the cell viability, migration and invasion of HTR8/SVneo cells [23]. There are also studies that show that silencing JARID2 can inhibit the progression of bladder cancer [24]. Hence, we speculated that miR-155 regulates CD34$^+$ endothelial cells by mediating JARID2.

Here, we isolated EPCs from mouse lung tissue and evaluated the state of CD34$^+$ EPCs by overexpressing or fsressing miR-155, amplifying or silencing JARID2, and simultaneously overexpressing/amplifying or suppressing/silencing miR-155 and JARID2. Finally, we assessed the effect of EPCs transplantation via the tail vein on CLG in a pneumonectomy model mouse.

## 2 Materials and methods

### 2.1 Separation and identification of EPCs

After the mice were anesthetized by intraperitoneal injection of 120–400 mg/kg Avertin, the lungs were excised in the airways to reduce additional lung airways, with the lung tissue cut into 1 cubic centimeter pieces and treated with collagenase (1 mg/mL; Sigma-Aldrich, St. Louis, MO, USA) and dispersed enzyme solution (2.5 U/mL; Sigma-Aldrich, St. Louis, MO, USA). The methods of isolating cells were performed according to Van Beijnum et al. [25] with minor modifications. Briefly, mononuclear cells were isolated by Ficoll gradient centrifugation (GE Healthcare, Piscataway, NJ, USA). Cells screened for CD34$^+$ with EasySep Immunomagnetic Bead Human CD34 Cell Positive Selection Reagent (STEMCELL Technologies,

Vancouver, BC, Canada) were seeded with fibronectin (10 μg/ml, BD Biosciences, San Jose, CA, USA) in EGM-2 medium (Lonza Group Ltd., Basel, Switzerland) containing 10% FBS. After 48 hours, the unadherent cells and cell fragments were removed and the adherent cell medium was replaced, and the fresh medium was replaced daily. After 7 days, adherent cells were screened based on EPC surface markers.

A total of $1 \times 10^5$ well-grown cells up to five passages were incubated with 1% BSA diluted antibody CD34-FITC (100 μL/tube, BD Biosciences, San Jose, CA), incubated on ice for 40 min in the dark. PBS rinses, centrifugation discards the supernatant, resuspends with PBS and then analyzes on the flow cytometer. Cells within 5 passages in a good growth state were collected and mixed with DiI-AC-LDL (10 μg/mL; BT-902, Bioquote, York, UK), incubated at 37˚C for 1 hour, fixed with 4% paraformaldehyde, incubated for 1 hour at 37˚C, and washed 3 times with PBS and then 1 time with distilled water. DAPI (ab228549, Abcam, UK) was added to the stained specimen, 50% buffered glycerol was added to mount the specimen, and fluorescence confocal microscopy was performed (Leica Microsystems GmbH, Germany).

## 2.2 Vector construction and cell transfection

JARID2 expression vector (pc-JARID2) inserted with the JARID2-coding sequence (CDS) for JARID overexpression. small interfering RNA JARID2 (si-JARID2), miR-155 mimics, inhibitor, and the corresponding control were synthesized by GenePharma (Shanghai, China). Transfection was performed according to the Lipofectam 2000 transfection kit (11668–500, Invitrogen, USA).

## 2.3 CCK-8 analysis of cell proliferation

Cells ($2 \times 10^6$) inoculated into 96-well plates and an equal amount of PBS solution was added to each well, and after the cells grew up against the wall, the supernatant was discarded and new medium was added. The plates were incubated at 37˚C, 5% $CO_2$ for 2 day. After that, adding CCK-8 solution (Dojindo Laboratories company, Japan) to the sample and modulating the enzyme label, the absorbance was measured at 450 nm wavelength.

## 2.4 Transwell assay of cell migration

The cells were inoculated at the top of a polycarbonate Transwell chamber without matrix glue, and the cells were cultured in serum-free medium. We added serum-containing culture medium into the bottom chamber. Then stained the fixed cells with crystal violet (Beyotime Institute of Biotechnology, Haimen, China), and finally observed and counted microscopically.

## 2.5 Tube formation experiment

Matrigel (BD Biosciences, San Jose, CA) was seeded for up to 1 hour at 37˚C to form reconstituted basement membranes. Transfected EPCs ($2 \times 10^4$ cells /100 μL) were harvested, seeded on Matrigels, and incubated for 6 hours at 37˚C. We examined the lumen structure under an inverted light microscope (IRB20, Microscope World, Carlsbad, CA, USA).

## 2.6 RT-qPCR

TRIzol RNA extraction kit (15596–018, Invitrogen, Madison, MI, USA) is served by extracting the total RNA from the collected cells and tissues. Then, the first strand of cDNA was assembled using the total RNA of the sample as a template, and the cDNA obtained in is used as a template for qPCR amplification, and subsequent PCR process was completed using cDNA, as

well as U6 and GAPDH as the internal reference, according to the instructions of SYBR Green qPCR kit. Then, the expression levels of cells and tissues were calculated using the $2^{-\Delta\Delta CT}$ method.

## 2.7 Western blotting

Proteins were extracted using RIPA buffers, followed by BCA protein kits measures protein concentration. The protein was isolated by electrophoresis of sodium dodecyl sulphate-poly-acrylamide gel and transferred to PVDF membrane. Then the PVDF membrane was incubated with primary antibody (Abcam, UK) including anti-JARID2 (ab184152,1:10000), anti-VEGFA (ab52917,1:10000), anti-MMP-2 (ab92536, 1:2500), and anti-GAPDH (ab181602,1:10000) and a HRP-conjugated anti-goat anti-rabbit IgG H&L secondary antibody (ab205718, 1:2000). After incubation, the membranes were finally assessed for expression semi-quantitatively by enhanced chemiluminescence (ECL) (Millipore, Billerica, MA, U. S. A.) chromogenic and gel imaging.

## 2.8 Animal model construction

In this study, C57BL/6 8-week-old mouse (Vital River Laboratory, Beijing, China) were picked out for the experiment. Mice were anesthetized by intraperitoneal injection of 120–400 mg/kg Avertin (Sigma, St. Louis, MO). After the left chest of mice was cleaned with an electric razor and disinfected with iodor, a ventral anterior axillary incision was made vertically from the left axilla to the left rib margin. After finding the left lung through the incision, the hilum was transected with tweezers and microdissection scissors. The incision was closed with PDS sutures. The mice were observed for at least 48 hours after surgery, and buprenorphene (0.1 mg/kg) was given every 8 hours for analgesia to alleviate the pain caused by surgery.

Three days after the animal model was established, the mouse were randomly assigned to (A) NC group ($1 \times 10^6$ EPCs were simply transplanted by tail vein injection); (B) miR-155 mimic + pc-NC group (EPCs transfected with miR-155 mimic and negative control JARID2); (C) NC mimic + pc-JARID2 group (EPCs transfected with negative control miR-155 mimic and pc-JARID2); (D) miR-155 mimic + pc-JARID2 group (EPCs overexpressing both miR-155 and JARID2 were transfected). The mice were sacrificed by cervical dislocation 4 days (The most active lung growth spot) after transplantation, and right lung tissue was removed for subsequent experiments. Animal studies were reviewed and approved by the Ethics Committee of Yunnan Cancer Hospital (SLKYLX202177), and in accordance with the National Guide for the Care and Use of Laboratory Animals.

## 2.9 Dual-luciferase reporter assay

The wild type JARID2 vector (WT) was constructed by cloning the JARID2 3 ' -UTR fragment containing the miR-155 binding site into the vector. An empty vector control, mutant JARID2vector (MUT), was also prepared. The reporter gene plasmid JARID2 WT and JARID2 MUT together with the transcription factor expression plasmid miR-155 mimic and the negative control (NC mimic) were transfected (Lipofectamine; 2000Invitrogen, Carlsbad, CA, USA) two days later. A dual luciferase reporter assay kit was (Promega, USA) used for analysis.

## 2.10 Lung volumetric and morphometric analysis

The mice were euthanized 4 days after the EPCs were transplanted and a water replacement procedure (Scherle) was performed to measure the remaining right lung volume (normalized). The specimens were treated with 10% formalin and 70% ethanol and embedded in paraffin for

histological analysis. The lung specimens were stained with hematoxylin and eosin (H&E) on the fourth day and the sections were sectioned for morphometric analysis [26,27]. The counts of points and intersections (on the overlaid 42-point grid) on each lung field (greater than 40) were observed for each slice at appropriate multiples to determine parameters such as lung parenchyma volume, total number of alveoli, and septal surface area.

## 2.11 Immunofluorescence analysis

The previously embedded sections were dewaxed with xylene and hydrated in gradient ethanol. Epitope repair was then performed using a pressurized chamber at 120˚C (Decloaking Chamber, Biocare Medical, Pacheco, California) by citrate solution (Vector Laboratories, Burlingame, CA)-based implementation. The slides were cleaned with PBST and incubated in a closed solution for 30 minutes. Incubate overnight with primary antibody (Ki67, ERG) at 4˚C. The next day, the slides were cleaned with PBST and incubated in Alexa Fluor-coupled IgG H&L (ab450073, Abcam, UK). After DAPI re-dyeing, PBST was washed, dried and fixed.

The lung sections of each group were observed with a fluorescence confocal microscope, and four random fields of the entire right lung were sampled for cell counting at 10 times magnification for each sample. Image J for counting positive cells, and the proliferating cells were manually counted and labeled based on the double staining of ERG and Ki67. Total endothelial cell were used for normalization calculation.

## 2.12 Statistical analysis

SPSS 21.0 software (IBM Corporation, Armonk, NY, USA) was served analyze the data. Datas are given in mean±SD. In statistical comparison, students' t-test was used when there were only two groups of differences between groups. Moreover, one-way ANOVA was served for multiple groups. $P<0.05$ indicates statistical significance.

# 3 Results

## 3.1 Cultivation and identification of EPCs

EPCs clusters and adherent cells formed fusiform structures after 3 and 7 days of culture, respectively (Fig 1A). After observing the results of immunofluorescence, we knew that DiI-ac LDL (red) could be absorbed by EPCs, making EPCs pink (Fig 1B). After screening with CD34[+] immunomagnetic beads, flow cytometric analysis to determine the identity of the EPCs showed that the CD34 positivity of the isolated cells was 87.4%, indicating that the isolated monocytes were CD34[+] EPCs (Fig 1C).

## 3.2 Effect of miR-155 on EPCs proliferation, migration, and tube formation

After evaluation by RT-qPCR, the quantitative data showed that miR-155 was significantly upregulated in the mimics group and downregulated in the inhibitor group (Fig 2A). Moreover, the level of JARID2 showed a clear downward trend in the miR-155 mimics group and upregulated in the inhibitor group (Fig 2B). These results indicate that in EPCs, miR-155 mimics and inhibitors can increase or decrease the level of miR-155, respectively, and have the above-mentioned opposite regulatory trends for JARID2.

By analyzing the resulting data from CCK-8, Transwell, and tube-forming assays, we learned that the miR-155 mimic group showed increased viability, migration, and tube-forming ability of EPCs, while that miR-155 inhibitor group show the opposite result (Fig 2C–2E). Moreover, the ELISA and Western blot detection results for angiogenesis-related factors

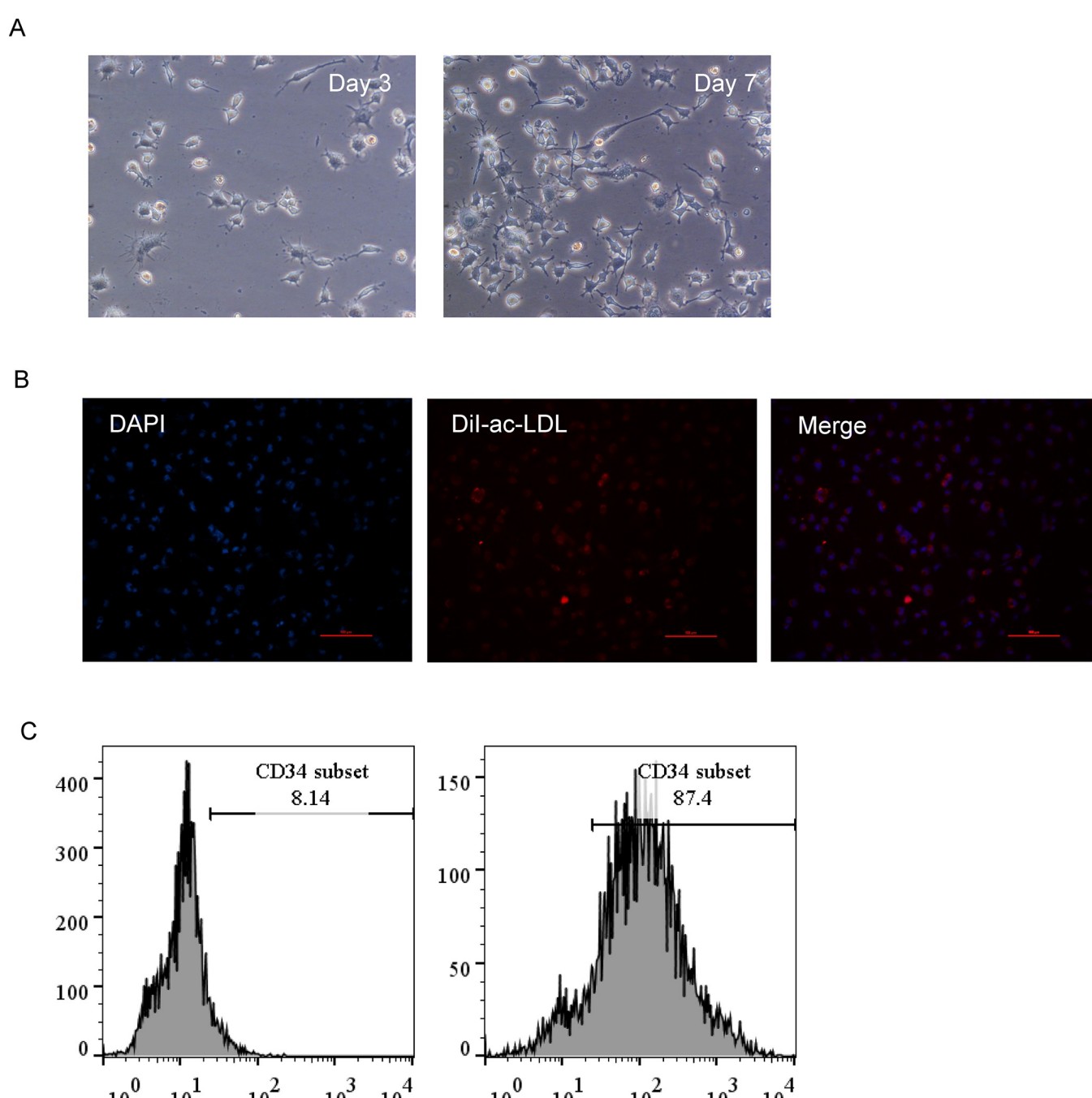

**Fig 1. Cultivation and identification of EPCs.** (A)EPCs morphology under an inverted light microscope (Days 3 and 7); (B) DiI-ac-LDL staining; (C)Flow cytometry analysis showing the positive expression of CD34 on EPCs and the positive expression of CD34 after immunomagnetic bead sorting.

showed that the secretion level of Ang-1 and the levels of VEGF and MMP-2 showed an upward trend in the mimics group but decreased in the inhibitor group (Fig 2F and 2G). It is known from that experimental data that upregulation of miR-155 can promote the development of EPCs, as well as the expression levels of the angiogenesis-related factors Ang-1, VEGF and MMP-2.

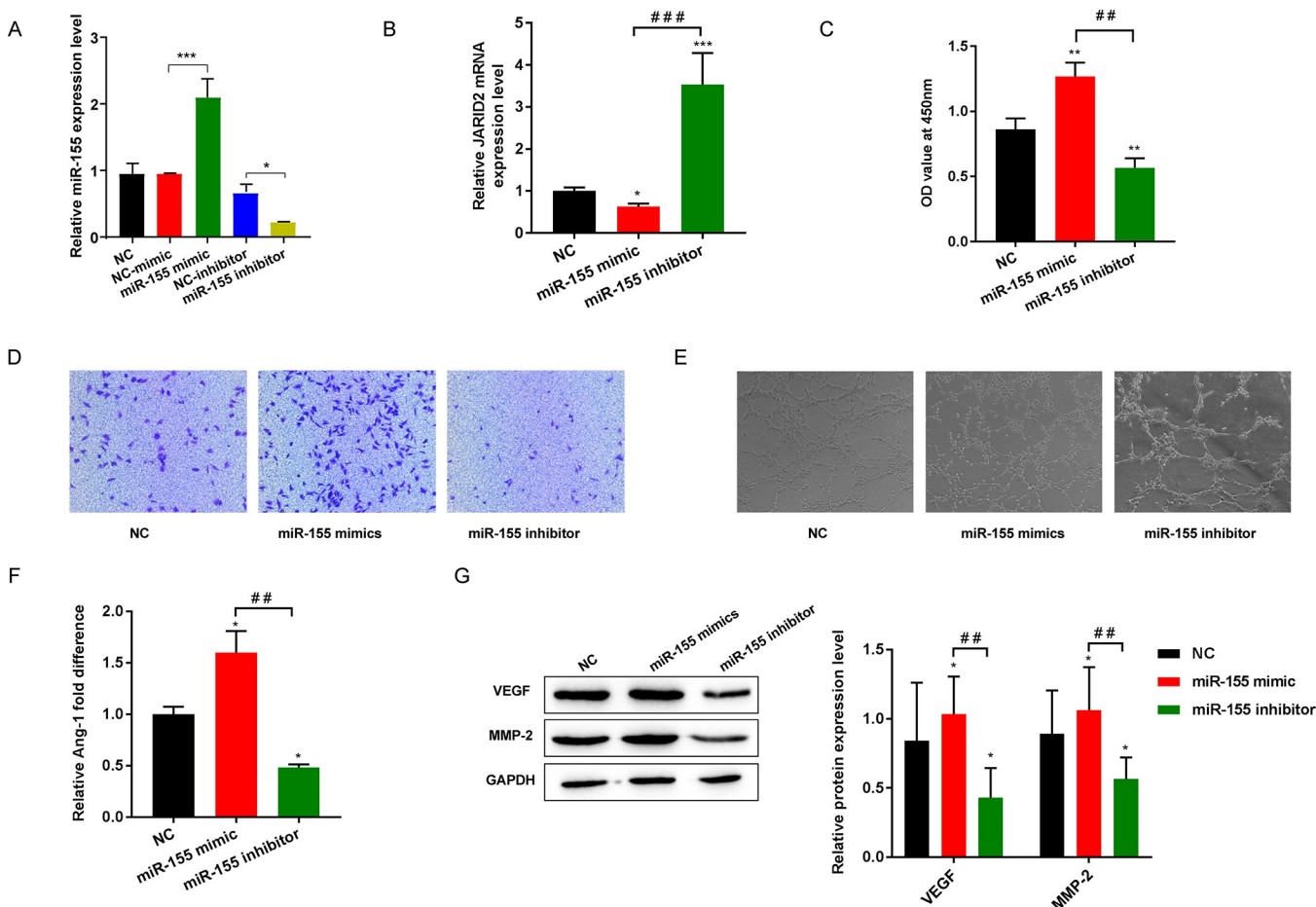

**Fig 2. Effect of miR-155 on EPCs proliferation, migration, and tube formation.** (A) The expression of miR-155 was detected by RT-qPCR; (B) The expression level of JARID2 mRNA was detected by RT-qPCR; (C) Cell proliferation activity was detected by CCK-8; (D) Cell migration level was detected by Transwell; (E) Detection of cell tube formation ability; (F) The level of Ang-1 secretion was detected by ELISA; (G) The expression levels of VEGF and MMP-2 protein were detected by Western blot. $*P<0.05$, $**P<0.01$, and $***P<0.001$ vs. NC group; $^{##}P<0.01$, and $^{###}P<0.001$.

### 3.3 Effect of JARID2 on EPCs proliferation, migration, and tube formation

EPCs were transfected with pc-JARID2 or si-JARID2 to observe the effect of JARID2. The negative controls (NC) was not transfected with pc-JARID2 or si-JARID2. We first used Western blotting to detect the protein expression level of JARID2 and found that JARID2 was markedly upregulated in the pc-JARID 2 group and downregulated in the si-JARID 2 group (Fig 3A). In addition, after evaluation by RT-qPCR, the quantitative data showed that miR-155 was significantly downregulated in the pc-JARID2 group, While in si-JARID2 group, the opposite trend was observed (Fig 3B). This finding suggests that pc-JARID2 can increase the level of JARID2 in EPCs while reducing the expression of miR-155. In contrast, si-JARID2 decreased JARID2 expression and increased miR-155 expression in EPCs.

By analyzing the resulting data from CCK-8, Transwell, and tube-forming assays, we learned that EPCs viability, migration, and tube formation were decreased in the pc-JARID2 group but increased in the si-JARID2 group (Fig 3C–3E). In addition, the ELISA and Western blot results of angiogenesis-related factors showed that the levels of Ang-1, VEGF and MMP-2 presented downward trend in the pc-JARID2 group but increased in the si-JARID2 group (Fig 3F and 3G). These results suggest that upregulation of JARID2 expression can inhibit the

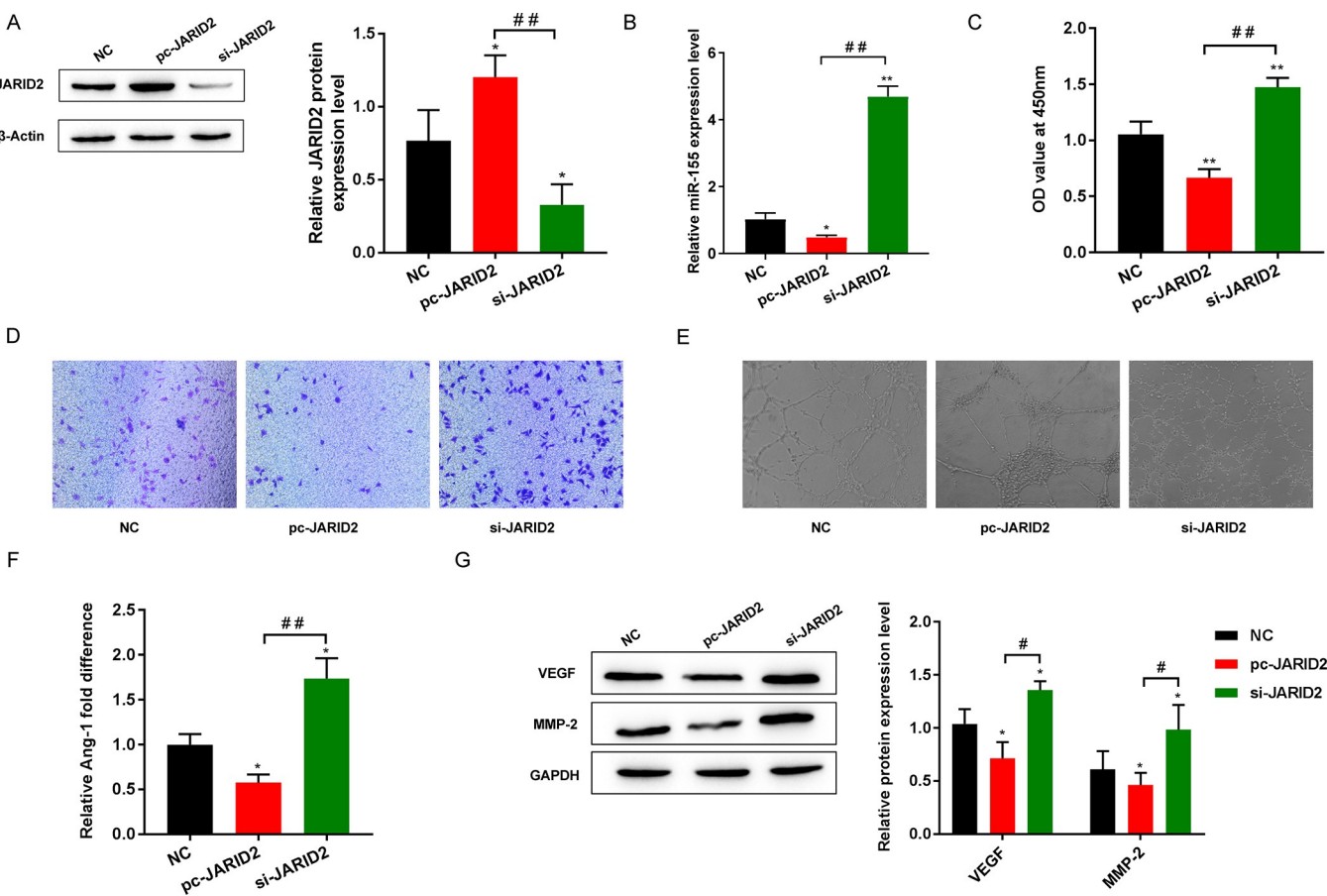

**Fig 3. Effect of JARID2 on EPCs proliferation, migration, and tube formation.** (A) The expression level of JARID2 protein was detected by Western blot; (B) The expression of miR-155 was detected by RT-qPCR; (C) CCK-8 to detect cell proliferation activity; (D) Cell migration level was detected by Transwell; (E) Detection of cell tube formation ability; (F) The level of Ang-1 secretion was detected by ELISA; (G) Western blot was used to detect the protein expression of VEGF and MMP-2. *$P<0.05$, **$P<0.01$ vs. NC group; #$P<0.05$, and ##$P<0.01$.

development of EPCs and the expression levels of the angiogenesis-related factors Ang-1, VEGF and MMP-2.

## 3.4 JARID2 was identified as a direct target of miR-155

We identified nine targets of miR-155 using StarBase (Fig 4A). We cotransfected JADIR2-WT/MUT and miR-155 mimic into 293T cells and found that the miR-155 mimic decreased the luciferase activity of JADIR2-WT in 293T cells but had no effects on JADIR2-MUT compared to the NC mimic group (Fig 4B). Inhibition of miR-155 markedly raised the level of JADIR2, while overexpression of miR-155 significantly reduced JADIR2 protein expression (Fig 4C). Altogether, it is known from that experimental data that miR-155 targeted JADIR2 genes.

## 3.5 Overexpression JARID2 weakened the promotion effect of miR-155 mimic on EPC proliferation, migration and tube formation

MiR-155 mimics and pc-JARID2 were simultaneously transfected into EPCs, with NC mimics and pc-NC as the respective negative controls. The negative controls (NC) was not transfected with miR-155 and JARID2. RT-qPCR results showed that transfection of miR-155 mimics

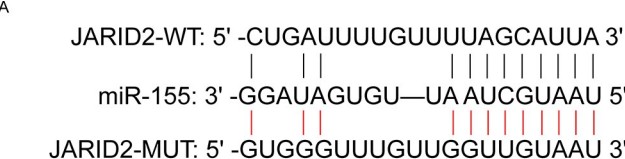

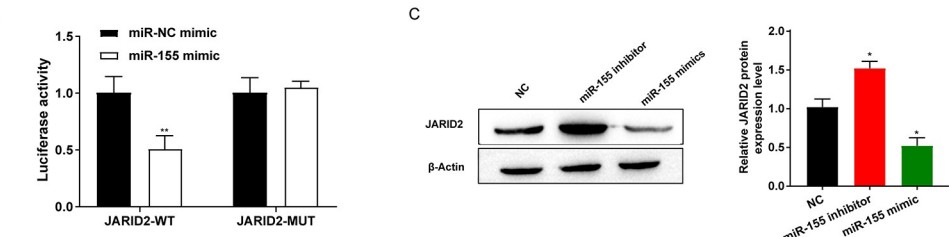

**Fig 4. JARID2 was identified as a direct target of miR-155.** (A) Sequence of binding sites of miR-155 and JARID2; (B) Dual luciferase reporter gene was used to detect the targeting relationship between miR-155 and JARID2; (C) JARID2 protein expression was detected by Western blot. $^{*}P<0.05$, $^{**}P<0.01$, vs. miR-NC mimic or NC group.

alone promoted the expression of miR-155 and inhibited the expression of JARID2, while transfection of pc-JARID2 alone had the opposite result. In addition, when miR-155 mimics was co-transfected with pc-JARID2, the effect of miR-155 mimics was weakened (Fig 5A and 5B).

By analyzing the resulting data from CCK-8, Transwell, and tube-forming assays, we learned that the miR-155 mimics+pc-NC group promoted the proliferative activity, migration, and tube formation of EPCs, while the NC mimics +pc-JARID2 group decreased the proliferative activity, migration, and tube formation of EPCs compared with the NC group. However, the miR-155 mimics+pc-JARID2 group reversed the effects of miR-155 mimics+pc-NC and NC mimics+pc-JARID2 groups on EPCs viability, migration, and tube formation (Fig 5C–5E). In addition, the results of ELISA and Western blot detection of angiogenesis related factors showed that compared with NC group, miR-155 mimics+pc-NC group increased the secretion level of Ang-1 and the levels of VEGF and MMP-2 in EPCs. NC mimics+pc-JARID2 decreased the secretion of Ang-1 and the levels of VEGF and MMP-2 in EPCs. However, the miR-155 mimics+pc-JARID2 group reversed the effects of the miR-155 mimics+pc-NC group and the NC mimics+pc-JARID2 group on Ang-1 secretion levels and VEGF and MMP-2 protein expression levels in EPCs (Fig 5F and 5G).

### 3.6 Silencing JARID2 weakened the inhibitory effect of miR-155 inhibitor on EPC proliferation, migration and tube formation

MiR-155 inhibitor and si-JARID2 were simultaneously transfected into EPCs, with NC inhibitor and si-NC as respective negative controls. The negative controls (NC) was not transfected with miR-155 and JARID2. After evaluation by RT-qPCR, the quantitative data showed that the level of miR-155 in the miR-155 inhibitor + si-NC group showed a lower trend than that in the NC group, but the JARID2 mRNA was higher. The level of miR-155 showed a clear upward trend in the NC inhibitor+si-JARID2 group, but the JARID2 mRNA level decreased. The level of miR-155 and JARID2 in the miR-155 inhibitor+si-NC group and NC inhibitor + si-JARID2 group were reverse in that miR-155 inhibitor+si-JARID2 group effect of expression level (Fig 6A and 6B).

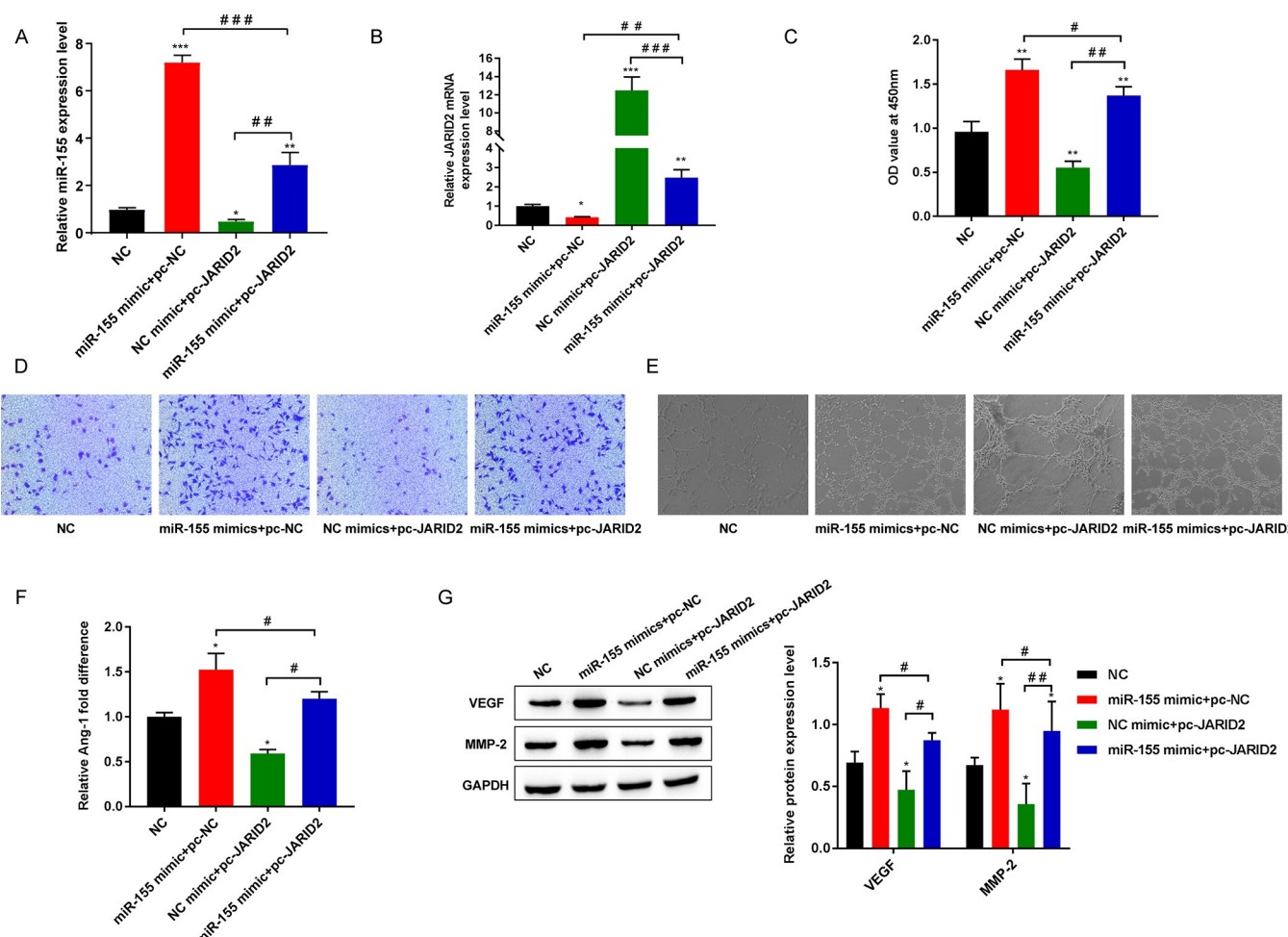

**Fig 5. Overexpression JARID2 weakened the promotion effect of miR-155 mimic on EPC proliferation, migration and tube formation.** (A) The expression of miR-155 was detected by RT-qPCR; (B) The expression level of JARID2 mRNA was detected by RT-qPCR; (C)CCK-8 was used to detect the cell proliferation activity; (D) Transwell detection of cell migration level; (E)Detection of cell tube forming ability; (F) Detection of the secretion level of Ang-1 by ELISA; (G) Western blot was used to detect the protein expression of VEGF and MMP-2. *$P<0.05$, **$P<0.01$, and ***$P<0.001$ vs. NC group; #$P<0.05$, ##$P<0.01$, and ###$P<0.001$.

We further revealed that miR-155 affects EPCs viability, migration, and tube formation through JARID2. The CCK-8, Transwell and tube formation assays showed that compared with the NC group, the miR-155 inhibitor +si-NC group showed reduced EPCs activity, migration and tube formation ability and the NC inhibitor + si-JARID2 group showed increased EPCs activity, migration and tube formation capacity. The miR-155 inhibitor + si-JARID2 group reversed the effects of the miR-155 inhibitor+si-NC group and the NC inhibitor + si-JARID2 group on EPCs viability, migration, and tube formation ability (Fig 6C–6E). In addition, the results of ELISA and Western blot detection of angiogenesis related factors showed that compared with the NC group, the miR-155 inhibitor+si-NC group reduced the secretion level of Ang-1 and the levels of VEGF and MMP-2 in EPCs. The NC inhibitor+si-JARID2 group increased the secretion of Ang-1 and expression of VEGF and MMP-2 proteins in EPCs. However, the miR-155 inhibitor+si-JARID2 group reversed the effects of the miR-155 inhibitor+si-NC group and the NC inhibitor+si-JARID2 group on the secretion level of Ang-1 and the levels of VEGF and MMP-2 in EPCs (Fig 6F and 6G).

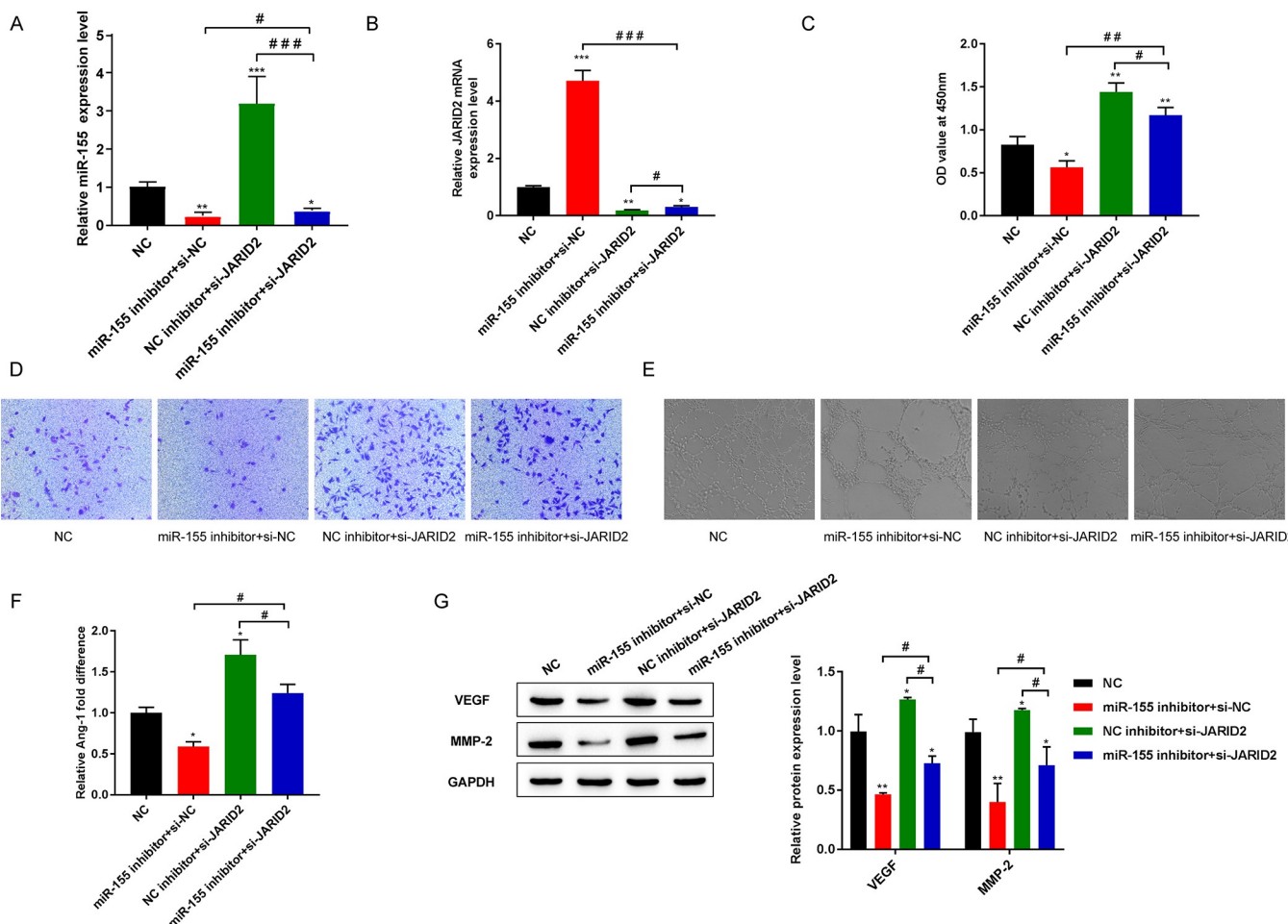

**Fig 6. Silencing JARID2 weakened the inhibitory effect of miR-155 inhibitor on EPC proliferation, migration and tube formation.** (A) The expression of miR-155 was detected by RT-qPCR; (B) The expression level of JARID2 mRNA was detected by RT-qPCR; (C) CCK-8 was used to detect the cell proliferation activity; (D) Transwell detection of cell migration level; (E) Detection of cell tube forming ability; (F) Detection of the secretion level of Ang-1 by ELISA; (G) Western blot was used to detect the protein expression of VEGF and MMP-2. *$P<0.05$, **$P<0.01$, and ***$P<0.001$ vs. NC group; #$P<0.05$, ##$P<0.01$, and ###$P<0.001$.

### 3.7 Effect of EPCs transplantation on parenchymal lung volume, diaphragm surface area, and alveolar number in mice

To assess the effect of EPCs transplantation on the lung histomorphology, we transfected EPCs with miR-155 mimics and pc-JARID2 simultaneously, NC mimics and pc-NC were negative controls for each, and NC was EPCs without transfection with miR-155 mimics or pc-JARID2. EPCs from each group were transplanted into mice by tail vein injection, and the morphology of lung tissue was analyzed after 4 days. Compared with the NC group, the lung parenchyma volume, the septal surface area and the total number of alveoli were significantly increased in the miR-155 mimics+pc-NC EPCs group and decreased in the NC mimics+pc-JARID2 EPCs group. However, the miR-155 mimics+pc-JARID2 in the EPCs group reversed this phenomenon (Fig 7A–7C).

### 3.8 Effect of EPCs transplantation on the number of pulmonary endothelial cells in mice

To further evaluate the effect of EPC transplantation on pulmonary endothelial cell proliferation. The results showed that compared with the control group, the number of endothelial

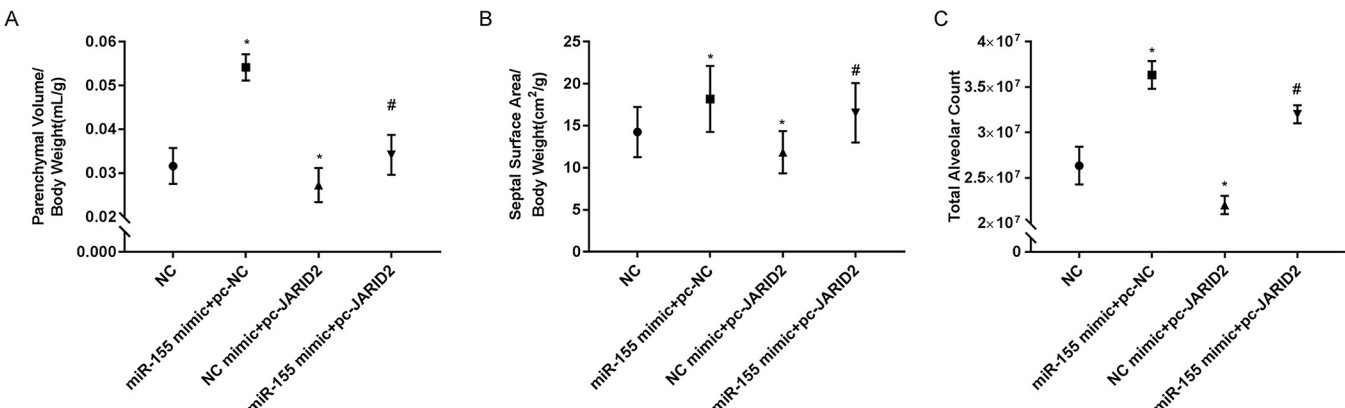

**Fig 7. Effect of EPCs transplantation on parenchymal lung volume, diaphragm surface area, and alveolar number in mice.** (A) Mouse parenchymal lung volume; (B) Diaphragm surface area; (C) Total alveolar count. *$P<0.05$ vs. NC group; #$P<0.05$ compared with the transfection group.

cells increased in the miR-155 mimics+pc-NC EPC group, decreased in the NC mimics+pc-JARID2 EPC group, and reversed this phenomenon in the miR-155 mimics+pc-JARID2 EPC group (Fig 8A and 8B).

## 4 Discussion

CLG is a confusing mechanism[28]. In humans, recent studies have shown that CLG may occur after pneumonectomy, although the timing may be several months to several years [29]. Compared with humans, pneumonectomy in small experimental animals results in CLG with complete recovery of lung volume [30]. A key factor in this regeneration is pulmonary angiogenesis during the formation of new alveoli. It is well known that EPCs has unique functions

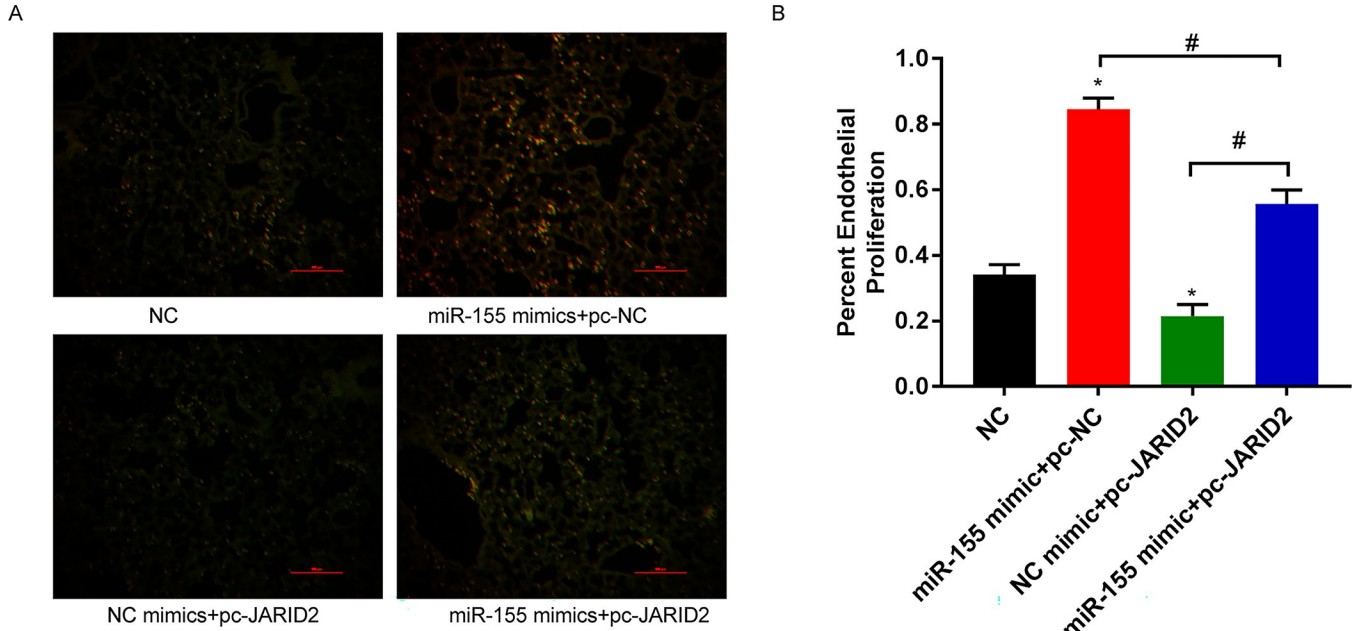

**Fig 8. Effect of EPCs transplantation on the number of pulmonary endothelial cells in mice.** (A) Partial representative lung immunofluorescence; (B) Percentage of endothelial cell proliferation. *$P<0.05$, compared with NC group; #$P<0.05$.

in angiogenesis as precursors of endothelial cells (ECs) [31,32]. These cells have the ability to differentiate and secrete cytokines to participate in angiogenesis [14]. In addition, CD34[+] EPCs have been shown to contribute to pulmonary angiogenesis and transform into resident ECs during CLG [10]. Therefore, the results of all these studies point to a possible mechanism between CD34[+] endothelial progenitor cells and lung compensatory growth. In this study, we found that EPCs transplantation increased the parenchymal lung volume, total alveolar number, septal surface area and the number of pulmonary endothelial cells, thereby promoting CLG after pneumonectomy in mice. These results indicate that activation of the biological function of EPCs can promote CLG. To explore the molecular mechanisms regulating the biological functions of EPCs, we carried out further studies.

MicroRNAs are small noncoding RNAs that they can silence or degrade mRNA, and then metabolize and degrade mRNA in biological processes, regulation of death. Moreover, they have unique functions in the regulation of cell migration, proliferation, apoptosis, and tube formation [33]. According to a large number of literatures, MiR-155 is a newly discovered miRNA, which has biological activities including lymphocyte activation, immune cell regulation, and microglial stimulation [34,35], and it is upregulated in a variety of diseases [36–39]. We also know from a recent report that miR-155 has a significant function in promoting tumor angiogenesis by inducing the downregulation of von Hippel-Lindau (VHL). Moreover, knockout of miR-155 inhibits physiological function of human umbilical vein endothelial cells (HUVECs) [40], as well as a similar situation was found in human retinal vascular endothelial cells (HRMECs) [41]. Hence, the content of this study is mainly to describe the biological behavior of the EPCs. It is found that up regulation of miR-155 may promote the EPCs. However, down-regulation of miR-155 significantly inhibited the physiological function of EPCs. These results indicate that the mechanism of action needs to be further investigated. Studies have shown that miR-155 targets JARID2 and has unique functions in cell migration and tube formation.

JARID2 is a DNA-binding Jumonji protein involved in the negative regulation of cell growth [42]. Silencing JARID2 promotes the proliferation and differentiation of normal CD34[+] cells[19], while amplification of JARID2 in HUVECs can lead to defective tube formation, indicating its inhibitory role in angiogenesis [43]. Furthermore, silencing JARID2 can enhance the viability of human cord blood and bone marrow-derived CD34[+] cells and act as a novel regulator with a critical role in hematopoietic stem and progenitor cell (HSPC) function [44]. However, the relationship between JARID2 and miR-155 has rarely been discussed. After summarizing and analyzing the experimental data, we learned that miR-155 can target JARID2 based on software predictions and literature studies. Then, We found that down-regulation of miR-155 can increase the level of JARID2, and the amplification of JARID2 can reverse the promotion of physiological activities of EPCs. Our study found that miR-155 affected the expression of JARID2 protein, and JARID2 also affected the expression of miR-155. Studies have shown that JARID2 can play a key role as a cofactor in the cross-talk between histone modification and PRC2 activity [45]. Therefore, we speculate that JARID2 may mediate the posttranslational modification of a certain protein, thereby indirectly affecting the expression of miR-155.

At present, there are few reports about miR-155 in CLG disease. Through the analysis of experimental results in vitro, it is known that miR-155 regulates EPCs proliferation, migration and tube formation through JARID2. Then, in vivo experiments found that miR-155 has a unique function in promoting CLG disease. In addition, H&E staining and immunofluorescence analysis showed that EPCs transfected with miR-155 increased the volume of lung parenchyma, the total number of alveoli and the surface area of septum, and significantly increased the number of pulmonary endothelial cells, while the simultaneous overexpression

of miR-155 and JARID2 reversed this trend. The possible mechanism of CLG is that miR-155 promotes CLG by regulating JARID2, and then affects the physiological function of EPCs.

## 5 Conclusion

In conclusion, our study reveals a novel role of miR-155 in the activation of EPCs and the promotion of CLG and provides important theoretical support for revealing the mechanism of puzzling CLG, which targets JARID2 to regulate EPCs proliferation, migration and tube formation.

## Supporting information

**S1 Raw images.**
(PDF)

**S1 Raw data.**
(ZIP)

**S2 Raw data.**
(ZIP)

**S3 Raw data.**
(ZIP)

**S4 Raw data.**
(ZIP)

**S5 Raw data.**
(ZIP)

**S6 Raw data.**
(ZIP)

**S7 Raw data.**
(ZIP)

**S8 Raw data.**
(ZIP)

## Author Contributions

**Conceptualization:** Li Zhao, Jing Peng.

**Data curation:** Li Zhuang, Zhiling Yan.

**Formal analysis:** Fei Liao.

**Funding acquisition:** Weiwei Wang.

**Investigation:** Li Zhao, Jing Peng.

**Methodology:** Yifan Wang, Shihao Shao.

**Project administration:** Weiwei Wang.

**Resources:** Weiwei Wang.

**Software:** Li Zhuang, Zhiling Yan.

**Supervision:** Weiwei Wang.

**Validation:** Li Zhao.

**Visualization:** Jing Peng.

**Writing – original draft:** Li Zhao, Jing Peng.

**Writing – review & editing:** Weiwei Wang.

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
