## [Decision Letter · Decision Letter 0]

7 Aug 2023

PONE-D-23-15191MiR-155 promotes compensatory lung growth by inhibiting JARID2 activation of CD34+ endothelial progenitor cellsPLOS ONE

Dear Dr. Wang,

Thank you for submitting your manuscript to PLOS ONE. After careful consideration, we feel that it has merit but does not fully meet PLOS ONE’s publication criteria as it currently stands. Therefore, we invite you to submit a revised version of the manuscript that addresses the points raised during the review process.

We look forward to receiving your revised manuscript.

Kind regards,

Zhijie Xu

Academic Editor

PLOS ONE

2. To comply with PLOS ONE submissions requirements, in your Methods section, please provide additional information regarding the experiments involving animals and ensure you have included details on (1) methods of sacrifice, and (2) efforts to alleviate suffering.

“This study was supported by the Regional Program of the National Natural Science Foundation of China (No. 81760423); Joint Special Fund for Applied Basic Research of Yunnan Provincial Science and Technology Department-Kunming Medical University (No. 202101AY070001-170).”

Reviewers' comments:

Reviewer's Responses to Questions

**Comments to the Author**

1. Is the manuscript technically sound, and do the data support the conclusions?

Reviewer #1: Yes

Reviewer #2: Yes

2. Has the statistical analysis been performed appropriately and rigorously? 

Reviewer #1: Yes

Reviewer #2: Yes

3. Have the authors made all data underlying the findings in their manuscript fully available?

Reviewer #1: Yes

Reviewer #2: Yes

4. Is the manuscript presented in an intelligible fashion and written in standard English?

Reviewer #1: Yes

Reviewer #2: Yes

5. Review Comments to the Author

Reviewer #1: This article tells a very interesting story and has a clear scientific research idea. However, it still needs to be revised before publication. Specific comments are as follows:

1. The part of animal experiments needs to be supplemented.

2.In animal modeling experiments, the basis for selecting 4 days as the time point needs to be supplemented.

3.Negative control groups were added to the mimics and inhibitors when the transfection efficiency was detected.

4.It is necessary to increase the citation of literature in recent years.

Reviewer #2: Manuscript submitted by Weiwei Wang “MiR-155 promotes compensatory growth by inhibiting JARID2 activation of CD34+ endothelial progenitor cells” convincingly showed in vitro (cell-culture) and in-vivo (mice model) that miR-155 controls activation of CD34-endothelial progenitor cell that lid to neovascularization and pulmonary angiogenesis (compensatory lung growth (CLG)) by the direct targeting of JARID2.

The results presented on the figure 3, 5 and 6 showing that not only that amount of miR-155 effected expression of JARID2 protein but also amount of JARID2 protein effect expression level of miR-155. It is not clear for me why authors did not discuss the effect of JARID2 protein on the expression level of miR-155 in the manuscript

The titles of two chapters of result section looks confusing for me:

274 3.5 Amplification of JARID2 Reverses the Effect of miR-155 on EPCs

275 Proliferation, Migration, and Tube Formation

300 3.6 Silencing JARID2 Reverses the Effect of miR-155 on EPCs Proliferation,

301 Migration, and Tube Formation

I think authors should specialized which effect of miR-155 was reversed by amplification and which one was reversed by silencing of JARID2.

In conclusion I recommend the manuscript for publication at PlOS1 journal after authors will addressed my concerns.

6. PLOS authors have the option to publish the peer review history of their article (what does this mean?). If published, this will include your full peer review and any attached files.

Reviewer #1: No

Reviewer #2: No

---

## [Author Response · Author response to Decision Letter 0]

11 Oct 2023

Dear Editors and Reviewers:

Thank you for your letter and for the reviewer's comments concerning our manuscript entitled "MiR-155 promotes compensatory lung growth by inhibiting JARID2 activation of CD34+ endothelial progenitor cells". Those comments are all valuable and very helpful for revising and improving our paper, as well as the important guiding significance to our researches. We have studied comments carefully and have made correction which we hope meet with approval. Revised portion are marked in red in the paper. The main corrections in the paper and the responds to the reviewer's comments are as flowing:

Responds to the journal requirements:

1. Comment: Please ensure that your manuscript meets PLOS ONE's style requirements, including those for file naming. The PLOS ONE style templates can be found at

Response: I have checked PLOS ONE style templates according to the link you provided, and revised our manuscript one by one according to the style samples.

Changes in the text: Level Heading, Figure Citations.

2. Comment: To comply with PLOS ONE submissions requirements, in your Methods section, please provide additional information regarding the experiments involving animals and ensure you have included details on (1) methods of sacrifice, and (2) efforts to alleviate suffering.

Response: According to your request, I have added the methods of animal sacrifice and the efforts made to alleviate their suffering in the methods section.

Changes in the text: Page 9, lines 171-173, 180.

3. Comment: Please also state specifically the method of anesthesia.

Response: I have added specific anesthesia methods in the section of Materials and methods according to your requirements.

Changes in the text: Page 5, lines 94-95; Page 8, lines 166-167.

4. Comment: Thank you for stating the following financial disclosure: “This study was supported by the Regional Program of the National Natural Science Foundation of China (No. 81760423); Joint Special Fund for Applied Basic Research of Yunnan Provincial Science and Technology Department-Kunming Medical University(No. 202101AY070001-170).” Please state what role the funders took in the study. If the funders had no role, please state: "The funders had no role in study design, data collection and analysis, decision to publish, or preparation of the manuscript." If this statement is not correct you must amend it as needed. Please include this amended Role of Funder statement in your cover letter; we will change the online submission form on your behalf.

Response: The funder is Weiwei Wang, the corresponding author of the fund applicant in this article. Weiwei Wang's role in the research has been elaborated in the author's contribution section, mainly in designing article ideas, reading and revising manuscripts.

Changes in the text: No changes are made in the text.

5. Comment: In your Data Availability statement, you have not specified where the minimal data set underlying the results described in your manuscript can be found. PLOS defines a study's minimal data set as the underlying data used to reach the conclusions drawn in the manuscript and any additional data required to replicate the reported study findings in their entirety. All PLOS journals require that the minimal data set be made fully available. For more information about our data policy, please see http://journals.plos.org/plosone/s/data-availability.

Response: I've sorted out all the raw data for the article, but I'm not sure how to get it into the database. So I uploaded the original data at the " Supporting Information - Compressed/ZIP File Archive " position in the submission system.

Changes in the text: No changes are made in the text.

6. Comment: Your ethics statement should only appear in the Methods section of your manuscript. If your ethics statement is written in any section besides the Methods, please move it to the Methods section and delete it from any other section. Please ensure that your ethics statement is included in your manuscript, as the ethics statement entered into the online submission form will not be published alongside your manuscript.

Response: I've moved the ethics statement to the methods section as you requested.

Changes in the text: Page 9, lines 182-184.

7. Comment: PLOS ONE now requires that authors provide the original uncropped and unadjusted images underlying all blot or gel results reported in a submission’s figures or Supporting Information files. This policy and the journal’s other requirements for blot/gel reporting and figure preparation are described in detail at https://journals.plos.org/plosone/s/figures#loc-blot-and-gel-reporting-requirements and https://journals.plos.org/plosone/s/figures#loc-preparing-figures-from-image-files. When you submit your revised manuscript, please ensure that your figures adhere fully to these guidelines and provide the original underlying images for all blot or gel data reported in your submission. See the following link for instructions on providing the original image data: 

https://journals.plos.org/plosone/s/figures#loc-original-images-for-blots-and-gels.

Response: I have sorted out all the uncropped blot/gel images in the manuscript according to your requirements and uploaded them through the system named file " S1_raw_images".

Changes in the text: No changes are made in the text.

Responds to the reviewer’s comments:

Reviewer 1

1. Comment: The part of animal experiments needs to be supplemented.

Response: Thanks for your comment on our paper. I have supplemented the content of animal experiments in more detail according to your request.

Changes in the text: Page 8, lines 166-170; Page 9, lines 171-173.

2. Comment: In animal modeling experiments, the basis for selecting 4 days as the time point needs to be supplemented.

Response: I have added in the manuscript that the basis for choosing 4 days as the time point is because this is the most active lung growth point.

Changes in the text: Page 9, lines 180-181.

3. Comment: Negative control groups were added to the mimics and inhibitors when the transfection efficiency was detected.

Response: Taking into account your comments, we have added a negative control group of mimics and inhibitors in the transfection efficiency detection experiment of miR-155 mimics and inhibitors.

Changes in the text: Figure 2A.

4. Comment: It is necessary to increase the citation of literature in recent years.

Response: Considering the reviewer's suggestion, I have replaced and updated some of the literature in the manuscript with some recent literature.

Changes in the text: Page 22, lines 444-445,448-452,455-466,469-470,484; Page 23, lines 485, 509-514.

Reviewer 2

1. Comment: The results presented on the figure 3, 5 and 6 showing that not only that amount of miR-155 effected expression of JARID2 protein but also amount of JARID2 protein effect expression level of miR-155. It is not clear for me why authors did not discuss the effect of JARID2 protein on the expression level of miR-155 in the manuscript.

Response: Taking into account your comment, we have supplemented the discussion of this result in the manuscript. Reviewing the literature, we learned that JARID2 can be used as a cofactor to mediate histone modification, so we speculate that JARID2 may mediate the post-translational modification of a protein and thus indirectly affect the expression of miR-155.

Changes in the text: Page 19, lines 396-400; Page 20, lines 401-404.

2. Comment: The titles of two chapters of result section looks confusing for me:

274 3.5 Amplification of JARID2 Reverses the Effect of miR-155 on EPCs

275 Proliferation, Migration, and Tube Formation

300 3.6 Silencing JARID2 Reverses the Effect of miR-155 on EPCs Proliferation,

301 Migration, and Tube Formation

Response: The contents of these two sections are actually intended to show that miR-155 affects EPC proliferation, migration and tube formation ability through JARID2. Therefore, we designed the experiment of overexpression and silence of JARID2 to observe the influence of JARID2 protein changes on the action of miR-155. It may be true that the expression of these two subheadings is not so clear that you are confused, so we have re-written these two subheadings as follows:

3.5 Overexpression JARID2 weakened the promotion effect of miR-155 mimic on EPC proliferation, migration and tube formation

3.6 Silencing JARID2 weakened the inhibitory effect of miR-155 inhibitor on EPC proliferation, migration and tube formation

Changes in the text: Page 14, lines 278-280; Page 15, lines 302-304.

3. Comment: I think authors should specialized which effect of miR-155 was reversed by amplification and which one was reversed by silencing of JARID2.

Response: Our results have shown that both overexpression and silencing of JARID2 affect the effects of miR-155 on EPC proliferation, migration and tube formation. So we have to supplement the experiments that need to be studied in your opinion.

Changes in the text: No changes are made in the text.

---

## [Decision Letter · Decision Letter 1]

18 Dec 2023

MiR-155 promotes compensatory lung growth by inhibiting JARID2 activation of CD34+ endothelial progenitor cells

PONE-D-23-15191R1

Dear Dr. Wang,

We’re pleased to inform you that your manuscript has been judged scientifically suitable for publication and will be formally accepted for publication once it meets all outstanding technical requirements.

Kind regards,

Zhijie Xu

Academic Editor

PLOS ONE

Additional Editor Comments (optional):

Reviewers' comments:

Reviewer's Responses to Questions

**Comments to the Author**

1. If the authors have adequately addressed your comments raised in a previous round of review and you feel that this manuscript is now acceptable for publication, you may indicate that here to bypass the “Comments to the Author” section, enter your conflict of interest statement in the “Confidential to Editor” section, and submit your "Accept" recommendation.

Reviewer #2: All comments have been addressed

2. Is the manuscript technically sound, and do the data support the conclusions?

Reviewer #2: Yes

3. Has the statistical analysis been performed appropriately and rigorously? 

Reviewer #2: Yes

4. Have the authors made all data underlying the findings in their manuscript fully available?

Reviewer #2: Yes

5. Is the manuscript presented in an intelligible fashion and written in standard English?

Reviewer #2: Yes

6. Review Comments to the Author

Reviewer #2: I am satisfied with author's manuscript corrections. My suggestions were addressed I recommend the manuscript for publication

7. PLOS authors have the option to publish the peer review history of their article (what does this mean?). If published, this will include your full peer review and any attached files.

Reviewer #2: No

---

## [Editor Report · Acceptance letter]

15 Feb 2024

PONE-D-23-15191R1 

PLOS ONE

Dear Dr. Wang, 

I'm pleased to inform you that your manuscript has been deemed suitable for publication in PLOS ONE. Congratulations! Your manuscript is now being handed over to our production team.

Kind regards, 

on behalf of

Prof. Zhijie Xu 

Academic Editor

PLOS ONE